# Allostatic Load Index Effectively Measures Chronic Stress Status in Zoo-Housed Giraffes

**Haley N. Beer [1,*], Lisa K. Karr [1], Trenton C. Shrader [2] and Dustin T. Yates [1]**

[1]  Department of Animal Science, University of Nebraska-Lincoln, Lincoln, NE 68503, USA;
     dustin.yates@unl.edu (D.T.Y.)
[2]  Department of Clinical Sciences, Kansas State University, Manhattan, KS 66506, USA
[*]  Correspondence: hbeer2@unl.edu

**Abstract:** For giraffes, few standardized methods exist for quantifying chronic stress. Allostatic load index is quantified from a panel of multi-system stress biomarkers to estimate cumulative stress. Our objective was to determine whether a panel of biomarkers selected for their role in allostatic load would reflect the number of documented stress events experienced by giraffes. Cortisol, DHEA-S, cholesterol, non-esterified fatty acids (NEFA), and fructosamine were determined in serum samples from zoo-housed giraffes (n = 18). These were correlated with the overall number and frequency of ZIMS-documented stress events experienced prior to blood collection. We also compared giraffes grouped by high vs. low total stress events and event frequencies. Giraffes experiencing higher total stress events tended to have 1.65-fold greater ($p < 0.10$) serum cortisol, had 1.49-fold greater ($p < 0.05$) serum fructosamine, and had 3.9-fold greater ($p < 0.05$) allostatic load. Giraffes experiencing higher stress-event frequency had 2.4-fold greater ($p < 0.05$) serum NEFA. Correlations for individual biomarkers with individual stress event categories were inconsistent, but DHEA-S (r = −0.44), cortisol/DHEA-S (r = 0.49), fructosamine (r = 0.54), and allostatic load (r = 0.49) correlated ($p < 0.10$) with total stress events. These findings indicate that the allostatic load index robustly reflected total cumulative stress events for these giraffes and was more consistent than the individual biomarkers used for its calculation. We conclude that allostatic load index is a promising tool for assessing stress in captive giraffes, although revision of the biomarker panel is warranted.

**Keywords:** stressors; morbidity; morality; zoo welfare; conservation

## 1. Introduction

The ability to accurately measure stress in animals has long been of interest due to its influence on health and disease. Indeed, poor health has been linked to sustained or frequently-occurring stress events arising from an individual's environment, food availability, social standing, or other cues [1]. Following a stress event, autonomic and endocrine responses initiate allostasis, the collective physiological responses that return the animal to biological homeostasis to enhance long-term survival [2,3]. Chronic or frequent stress responses with inadequate recovery time between events can negatively impact physical and mental health in part by causing allostatic systems to fatigue [4]. Cellular components including enzymes, receptors, and substrate channels can become functionally compromised, which weakens allostatic mechanisms over time and reduces the ability to cope with subsequent stressors [5,6]. Such dysregulation due to cumulative stressful experiences is known as allostatic load [5,6]. Studies in humans show that high allostatic load indices are associated with development of chronic degenerative conditions, including compromised immune function, hypertension, and heart failure [7]. Many of the major pathophysiological mediators and outcomes are conserved across mammalian species [8]. Therefore, non-human mammals with elevated allostatic load are also presumably at greater risk for poor health and shorter life expectancy [4,8]. Predisposing demographic, genetic, environmental, and developmental factors can influence sensitivity to stress and, by proxy, the

rate at which allostatic load progresses [9]. By drawing upon the same approach used in humans, species-specific allostatic load indices can be formulated for zoo animals as well. This involves an initial assessment of a robust panel of candidate biomarkers encompassing a broad range of physiological processes that acknowledge and account for subtle species-specific considerations [4]. Typically, allostatic load is calculated as a comprehensive value that sums high-risk biomarkers (based on a predetermined percentile) into a single composite score [4]. Although originally applied to humans, this methodology has also shown potential for assessing stress and predicting health outcomes in animals [8,9]. We postulated that a panel for assessing allostatic load in captive giraffe populations could be established from blood serum components. Our selection of parameters from within—cortisol, dehydroepiandrosterone sulfate (DHEA-S)—and beyond—cholesterol, non-esterified fatty acids (NEFA), fructosamine—the hypothalamic-pituitary–adrenal (HPA) axis is informed by human literature and characteristics specific to giraffes. Cortisol has been extensively utilized as a stress indicator in animals and in some circumstances can identify greater risk for morbidity and mortality [6,10–18]. Circulating DHEA-S concentrations can often provide an indication of how chronically the HPA axis is being activated, as it can function as a prohormone for adrenal corticosteroids [19,20]. Elevated total cholesterol can be indicative of chronic nutritional stress and is also associated with systemic inflammation [21–25]. Likewise, circulating NEFA concentrations have been used as biomarkers of stress-induced cardiovascular, endocrine, and metabolic dysfunction [26–31]. In addition to the selected serum components, we incorporated body condition scores, which estimate physical fitness and can be predictive of an animal's ability to successfully cope with stress [32–39]. The accumulation of frequent or chronic stress events over time has been linked to greater morbidity and mortality rates in humans and animals [6,9,40,41] For comparisons of health within and among populations to be valid, a standardized panel of diagnostic parameters with known normal values must be established. Thus, the objective of this study was to evaluate parameters that potentially contribute to allostatic load and compare them to the known number and frequency of documented stress events in a population of zoo-housed giraffes (*Giraffa camelopardalis*).

## 2. Materials and Methods

### 2.1. Subjects

To evaluate biomarkers of chronic stress in captive giraffes (*Giraffa camelopardalis*), frozen serum samples were obtained from 18 individuals (6 male, 12 female) housed at Omaha's Henry Doorly Zoo and Aquarium (OHDZA) in Omaha, NE, USA. Animals averaged 12.8 years of age and ranged from 2 to 26 years. They were reported to be in good general health at the time of sample collection, per evaluations by animal care staff. All samples were from opportunistic blood draws via jugular venipuncture, and the most recent sample was used for animals with multiple samples in the repository. Samples were collected from 16 of the individuals between 2010 and 2012 and from the remaining two individuals in 2017 and 2018, respectively. All samples were collected via procedures that were approved by the OHDZA Institutional Care and Use Committee. Serum was separated by centrifugation, placed on ice, and stored at −80 °C until the summer of 2018.

### 2.2. Sample Collection

Blood was collected via jugular venipuncture using an 18G hypodermic needle and 20 mL syringe. Whole blood was transferred in serum separator tubes (Becton, Dickson & Co., Franklin Lakes, NJ, USA) and centrifuged at $2000 \times g$ for ten minutes. The supernatant was transferred via Pasteur pipette into cryogenic storage vials (ThermoFisher Scientific, Waltham, MA, USA) and stored at −80 °C. At the request of the researchers, samples were shipped overnight on ice to the Cornell Clinical Pathology Lab and the St. Louis Zoo Endocrinology Lab for analysis.

*2.3. Serum Analyses*

Serum cortisol and DHEA-S concentrations were determined by the St. Louis Endocrinology Laboratory as previously validated for ruminants [42]. Samples were assessed in duplicate with commercial ELISA kits (DetectX Cortisol K003-H5W and DetectX DHEA-S K054-H5, respectively; Arbor Assays, Ann Arbor, MI) with lower detection limits of 45.4 pg/mL and 75.6 pg/mL, respectively. Samples were diluted 1:100 for cortisol and 1:1000 for DHEA-S with assay buffer from Arbor Assays, as recommended by the manufacturer. The remaining serum components were determined by the Cornell University College of Veterinary Medicine Pathology Laboratory. Serum total cholesterol concentrations were evaluated using the CHOD-PAP colorimetric method as previously described [43]. Serum NEFA were measured via the lab's previously validated colorimetric enzymatic methods [44]. Likewise, serum fructosamine concentrations were determined using the lab's previously validated nitroblue tetrazolium assay [45].

*2.4. Body Condition Scoring*

Body condition scores (BCS) are species-specific metrics for describing fitness and apparent body composition based on visual evaluations [34,38,46]. BCS were recorded by zoo veterinarians for each giraffe at or near the time of blood collection. Values are presented on a standard 1–9 scale, with values corresponding to the animal's physical condition as below:

1 = Emaciated condition, absent visible fat, prominent skeletal details, muscle wasting.
2 = Intermediate between 1 and 3.
3 = Lean body condition, suboptimal fat stores, noticeable skeletal details, lean muscling.
4 = Intermediate between 3 and 5.
5 = Ideal condition, adequate fat, only notable features of skeletal frame (pelvis and spine).
6 = Intermediate between 5 and 7.
7 = Overweight condition, lost definition of notable features, fat deposition on chest.
8 = Intermediate between 7 and 9.
9 = Obese condition, extensive fat, palpable loss of skeletal features, protruding abdomen.

*2.5. Allostatic Load*

To create an allostatic load index for the giraffes, values for each of the 6 indicators (5 serum biomarkers and BCS) were determined to be within or outside of the normal range for the study population using the combination of quartile and 2-tailed deciles recently described for zoo-housed gorilla studies [47–49]. The allostatic load metric was calculated as the number of indicators (0–6) for which an abnormal value was recorded. Mean and median values for each indicator are shown in Table 1. As a starting point, abnormal values for this study were determined as the most extreme 25% of the population for that indicator. Thus, the highest 12.5% and lowest 12.5% of BCS and serum cholesterol values, the highest 24% of serum cortisol and NEFA values, and the lowest 25% of serum DHEA-S and fructosamine values were considered abnormal for the study population.

*2.6. Compilation of Stress Events*

Stress events were considered to be any immobilizations, transfers, illnesses, and pregnancies documented in the Zoo Information Management Software (ZIMS) digital medical records that occurred prior to the blood sample being collected. Tabulation of these stress events is summarized in Figure 1. This information was derived from the medical and husbandry records provided to ZIMS by the cooperating institution in August 2018. Immobilizations were defined as events during which an anesthetic was utilized to facilitate or accompany restraint. Transfers were defined as events during which the animal was loaded into a trailer and transported. Illnesses were defined as an explicit documented disease or injury that required medical intervention. This included the recurrence of a previously resolved issue. For example, lameness of a limb that was resolved to the satisfaction of the attending veterinarian but that subsequently presented as limb lameness several

months later would be defined as two separate stress events. Pregnancies were included when confirmed by observed birth or rectal ultrasound. These included pregnancies that resulted in stillbirth or abortion.

**Table 1.** Average values for stress biomarkers and occurrences of documented stress events in zoo-housed giraffes.

| Variable | N | Mean | Std. Error | Std. Dev. | Median | Minimum | Maximum |
|---|---|---|---|---|---|---|---|
| Age, years | 18 | 12.8 | 1.8 | 7.6 | 12.5 | 2 | 26 |
| BCS | 18 | 4.78 | 0.17 | 0.73 | 5 | 3 | 6 |
| Cortisol, mg/dL [1] | 18 | 35.8 | 6.3 | 26.7 | 25.6 | 6.8 | 99.6 |
| DHEA-S, mg/dL [1] | 18 | 46.7 | 6.4 | 27.3 | 53.3 | 5.4 | 89.4 |
| Cortisol/DHEA-S [1] | 18 | 1.77 | 0.76 | 3.21 | 0.71 | 0.1 | 13.3 |
| Cholesterol, mg/dL [1] | 18 | 29.7 | 3.3 | 14.1 | 31 | 4 | 54 |
| NEFA, mEq/L [1] | 18 | 0.31 | 0.05 | 0.20 | 0.23 | 0.07 | 0.68 |
| Fructosamine, mM [1] | 18 | 387 | 36 | 155 | 355 | 236 | 870 |
| Immobilizations [2] | 18 | 6.1 | 1.5 | 6.4 | 3 | 0 | 20 |
| Illnesses [2] | 18 | 4.8 | 0.5 | 2.2 | 4 | 2 | 9 |
| Pregnancies [2] | 18 | 0.72 | 0.23 | 0.96 | 0 | 0 | 3 |
| Transports [2] | 18 | 0.89 | 0.16 | 0.68 | 1 | 0 | 2 |
| Total Stress Events | 18 | 12.50 | 1.66 | 7.04 | 13.50 | 4 | 26 |
| Stress Events/Year | 18 | 1.07 | 0.13 | 0.57 | 0.92 | 0.3 | 2.2 |
| Allostatic Load | 18 | 1.44 | 0.29 | 1.25 | 1 | 0 | 5 |

[1] Circulating biomarkers were measured in serum samples collected by jugular venipuncture. [2] Stress events occurring prior to blood sample were documented by OHDZA staff in ZIMS. BCS, body condition score; DHEA-S, dehydroepiandrosterone sulfate; NEFA, non-esterified fatty acids; NS, not significant.

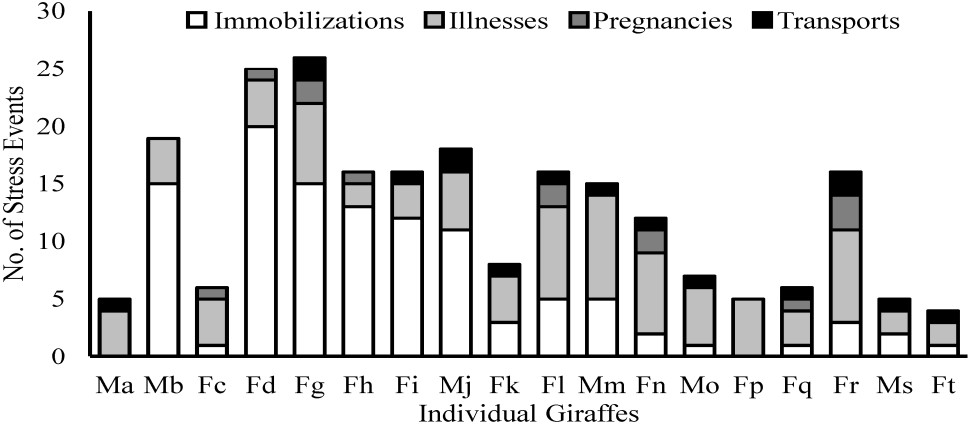

**Figure 1.** Stress events documented in ZIMS that were used to calculate allostatic load for each individual giraffe in a zoo-housed population. Uppercase letters denote sex (M, male; F, female) and lowercase letters represent the individual giraffe.

*2.7. Statistical Analysis*

Statistical analyses for the effects of total stress events (highest vs. lowest 50th percentile), stress event frequency (highest vs. lowest 50th percentile), age group (over vs. under 10 years of age), and sex (male vs. female) were performed by ANOVA using the MIXED procedure of SAS (SAS Institute, Cary, NC, USA). Giraffe was considered the experimental unit. The CORR procedure of SAS was used to determine Pearson and Spearman correlation coefficients between each individual biomarker and each stress event category, as well as total stress events, stress event frequency, and allostatic load. Correlation analyses presented in the main text were performed for the entire group of giraffes, but similar analyses were performed for males only, females only, over 10 years of age only, and under 10 years of age only, and these are included as Supplementary Materials. The threshold

for statistical significance was $p \leq 0.05$, and tendencies toward statistical significance were considered at $p \leq 0.10$. Data are presented as means $\pm$ standard error.

## 3. Results

Serum cortisol concentrations and cortisol/DHEA-S ratios tended to be greater ($p < 0.10$) and serum fructosamine concentrations were less ($p < 0.05$) for giraffes in the highest 50th percentile for total stress events than those in the lowest 50th percentile (Figure 2), but serum DHEA-S, cholesterol, NEFA did not differ between total stress event groups. Serum NEFA concentrations were greater ($p < 0.05$) for giraffes in the highest 50th percentile for stress event frequency than for those in the lowest 50th percentile (Figure 3), but serum cortisol, DHEA-S, cholesterol, and fructosamine concentrations did not differ between stress event frequency groups. Serum cortisol was less ($p < 0.05$), serum DHEA-S tended to be less ($p < 0.10$), and serum cholesterol was greater ($p < 0.05$) for giraffes 10 years of age or older than for those under 10 years of age (Figure 4), but serum NEFA and fructosamine concentrations did not differ between age groups. Serum cholesterol concentrations were less ($p < 0.05$) for female giraffes than for male giraffes (Figure 5), but serum cortisol, DHEA-S, NEFA, and fructosamine concentrations did not differ between sexes. Moreover, BCS did not differ between groups for total stress events, stress event frequency, age, or sex (Figure 6). Allostatic load indices were greater ($p < 0.05$) for giraffes in the highest 50th percentile for total stress events than those in the lowest 50th percentile (Figure 7), but did not differ between stress event frequency groups, age groups, or sexes.

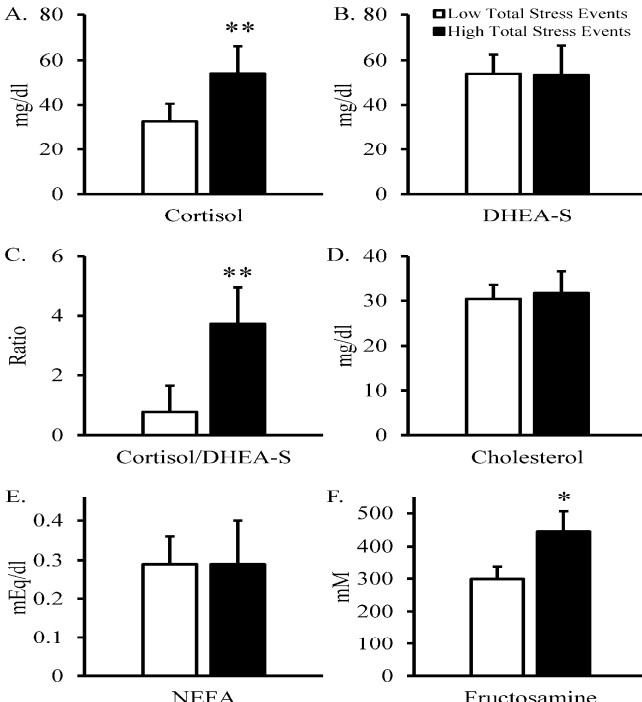

**Figure 2.** Stress biomarkers assessed in serum samples from zoo-housed giraffes grouped by total stress events experienced. Serum cortisol concentrations (**A**), DHEA-S concentrations (**B**), cortisol-to-DHEA-S ratios (**C**), cholesterol concentrations (**D**), NEFA concentrations (**E**), and fructosamine concentrations (**F**) were compared between giraffes in the highest (n = 9) and lowest (n = 9) 50th percentiles for total documented stress events (immobilizations, illnesses, pregnancies, transport events) prior to the time of blood sampling. Means that differed ($p < 0.05$) for a serum component are denoted by *. Means that tended to differ ($p < 0.10$) are denoted by **.

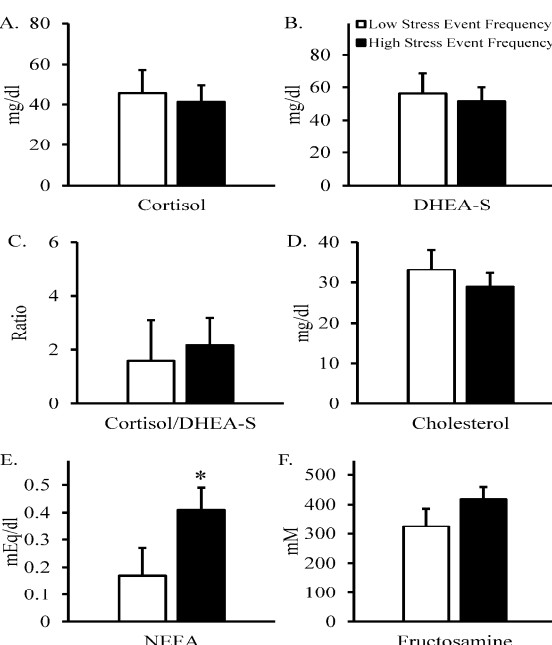

**Figure 3.** Stress biomarkers assessed in serum samples from zoo-housed giraffes grouped by stress event frequency. Serum cortisol concentrations (**A**), DHEA-S concentrations (**B**), cortisol-to-DHEA-S ratios (**C**), cholesterol concentrations (**D**), NEFA concentrations (**E**), and fructosamine concentrations (**F**) were compared between giraffes in the highest (n = 9) and lowest (n = 9) 50th percentiles for average documented stress events (immobilizations, illnesses, pregnancies, transport events) per year prior to the time of blood sampling. Means that differed (*p* < 0.05) for a serum component are denoted by *.

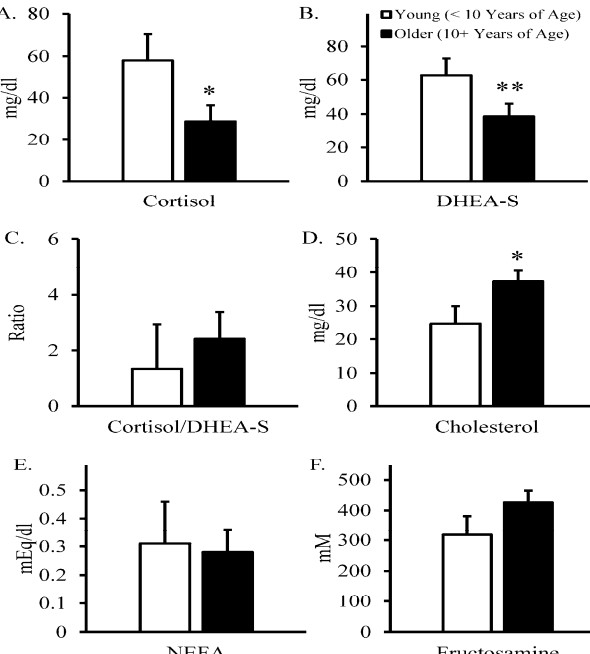

**Figure 4.** Stress biomarkers assessed in serum samples from zoo-housed giraffes grouped by age. Serum cortisol concentrations (**A**), DHEA-S concentrations (**B**), cortisol-to-DHEA-S ratios (**C**), cholesterol concentrations (**D**), NEFA concentrations (**E**), and fructosamine concentrations (**F**) were compared between giraffes that were under 10 years of age (n = 6) and those that were 10 years or older (n = 6) prior to the time of blood sampling. Means that differed (*p* < 0.05) for a serum component are denoted by *. Means that tended to differ (*p* < 0.10) are denoted by **.

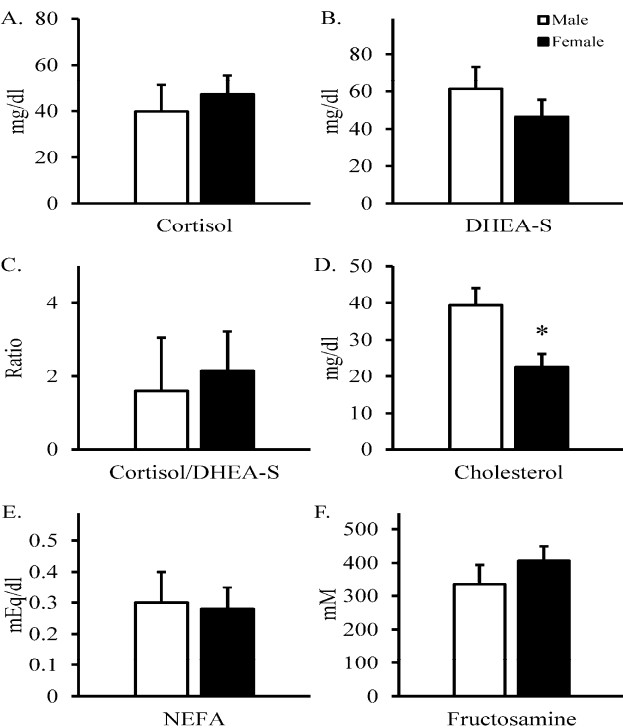

**Figure 5.** Stress biomarkers assessed in serum samples from zoo-housed giraffes grouped by sex. Serum cortisol concentrations (**A**), DHEA-S concentrations (**B**), cortisol-to-DHEA-S ratios (**C**), cholesterol concentrations (**D**), NEFA concentrations (**E**), and fructosamine concentrations (**F**) were compared between male (n = 6) and female (n = 12) giraffes. Means that differed (*p* < 0.05) for a serum component are denoted by *.

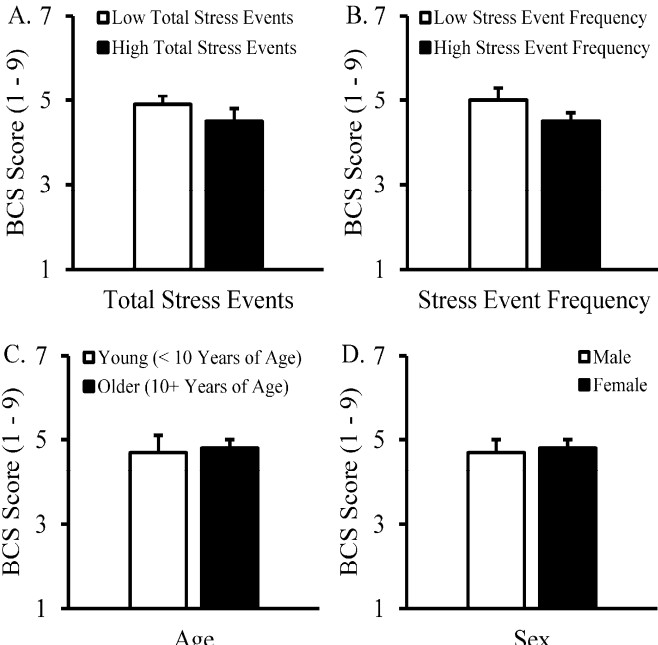

**Figure 6.** Body condition scores (BCS) in zoo–housed giraffes. Comparisons are shown for giraffes grouped by total stress events (**A**), stress event frequency (**B**), age (**C**), and sex (**D**). BCS were assigned using a 1–9 score based on standardized criteria.

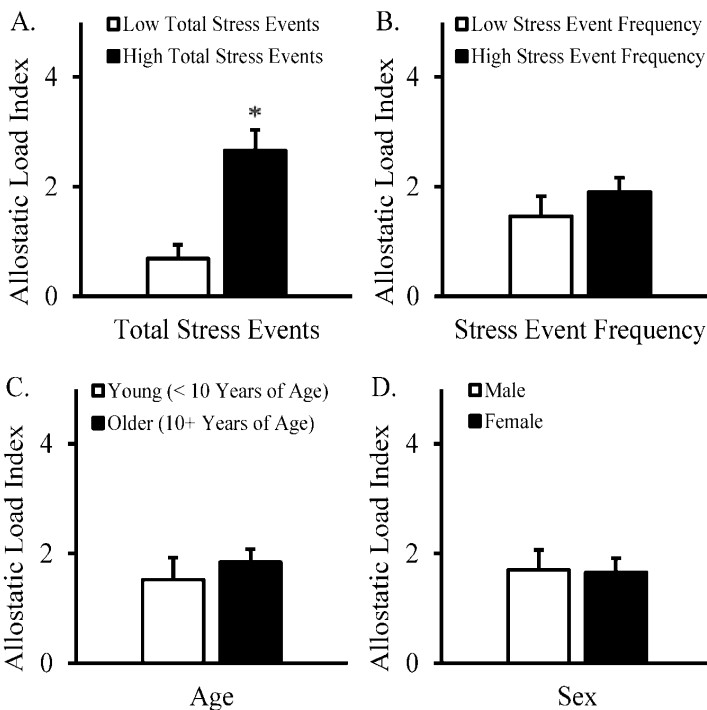

**Figure 7.** Allostatic load index in zoo-housed giraffes. Comparisons are shown for giraffes grouped by total stress events (**A**), stress event frequency (**B**), age (**C**), and sex (**D**). Allostatic load was calculated from outliers for serum cortisol, DHEA-S, cholesterol, NEFA, and fructosamine concentrations as well as body condition score. Outliers were identified using standard quartile and two-tailed decile criteria. Means that differed ($p < 0.05$) for a serum component are denoted by *.

Pearson correlation analyses indicated no correlations between serum cortisol and any individual stress event categories, total stress events, stress event frequency, or allostatic load (Table 2). Likewise, serum cholesterol and BCS did not correlate with any of these categories. Serum DHEA-S was negatively correlated with total illnesses and total pregnancies and tended to be negatively correlated ($p < 0.10$) with age and total stress events. Cortisol/DHEA-S ratios were positively correlated ($p < 0.05$) with allostatic load and total stress events and tended to be positively correlated ($p < 0.10$) with total illnesses and total transports. Serum NEFA tended to be negatively correlated ($p < 0.10$) with stress event frequency. Serum fructosamine was positively correlated ($p < 0.05$) with total immobilization and total stress events and tended to be positively correlated ($p < 0.10$) with allostatic load. Allostatic load was positively correlated ($p < 0.05$) with total stress events. Pearson correlation analyses performed for giraffes grouped by sex and age are presented in Supplemental Tables S1–S4.

Spearman correlation analyses indicated that BCS tended to be negatively correlated ($p < 0.10$) with allostatic load (Table 3). Serum cortisol tended to be negatively correlated ($p < 0.10$) with stress event frequency. Serum DHEA-S was negatively correlated ($p < 0.05$) with total illnesses and tended to be negatively correlated ($p < 0.10$) with age and total pregnancies. Spearman correlation analyses did not indicate correlations between cortisol/DHEA-S, serum cholesterol or NEFA, and any stress categories. Serum fructosamine was positively correlated with allostatic load and total immobilizations and tended to be positively correlated ($p < 0.10$) with age and total stress events. Allostatic load tended to be negatively correlated ($p < 0.10$) with BCS. Spearman correlation analyses performed for giraffes grouped by sex and age are presented in Supplemental Tables S5–S8.

**Table 2.** Pearson correlation coefficients [1] for stress biomarkers and occurrences of documented stress events in zoo-housed giraffes.

| Variable | Variable | | | | | | | | |
|---|---|---|---|---|---|---|---|---|---|
| | Age | BCS | Cortisol | DHEA-S | Cortisol/DHEA-S | Cholesterol | NEFA | Fructosamine | Allostatic Load |
| Age | | 0.29 | −0.26 | −0.41 | 0.28 | 0.35 | 0.00 | 0.19 | 0.16 |
| | | NS | NS | 0.09 | NS | NS | NS | NS | NS |
| BCS | 0.29 | | −0.06 | −0.23 | 0.11 | 0.06 | 0.24 | −0.09 | −0.21 |
| | NS | | NS | NS | NS | NS | NS | NS | NS |
| Allostatic Load | 0.16 | −0.21 | 0.35 | −0.36 | 0.79 | 0.38 | 0.25 | 0.45 | |
| | NS | NS | NS | NS | <0.01 | NS | NS | 0.06 | |
| Immobilizations | 0.56 | −0.12 | 0.05 | −0.14 | 0.31 | 0.30 | 0.27 | 0.53 | 0.36 |
| | 0.02 | NS | NS | NS | NS | NS | NS | 0.02 | NS |
| Illnesses | 0.42 | 0.26 | −0.23 | −0.69 | 0.43 | 0.15 | 0.23 | 0.16 | 0.36 |
| | 0.09 | NS | NS | <0.01 | 0.08 | NS | NS | NS | NS |
| Pregnancies | 0.58 | 0.33 | −0.05 | −0.50 | 0.25 | −0.19 | 0.11 | 0.24 | 0.16 |
| | 0.01 | NS | NS | 0.04 | NS | NS | NS | NS | NS |
| Transports | 0.12 | 0.30 | 0.12 | −0.27 | 0.41 | −0.02 | 0.37 | −0.24 | 0.34 |
| | NS | NS | NS | NS | 0.09 | NS | NS | NS | NS |
| Total Stress Events | 0.73 | 0.05 | −0.02 | −0.44 | 0.49 | 0.29 | 0.37 | 0.54 | 0.49 |
| | <0.01 | NS | NS | 0.07 | 0.04 | NS | NS | 0.02 | 0.04 |
| Stress Events/Year | 0.47 | 0.34 | −0.29 | 0.09 | −0.13 | 0.23 | −0.42 | −0.15 | −0.24 |
| | 0.05 | NS | NS | NS | NS | NS | 0.08 | NS | NS |

[1] Pearson correlation coefficients (r) are the top number for each cell. *p*-values for each correlation are italicized and below each coefficient. BCS, body condition score; DHEA-S, dehydro-epiandrosterone sulfate; NEFA, non-esterified fatty acids; NS, not significant.

**Table 3.** Spearman correlation coefficients [1] for stress biomarkers and occurrences of documented stress events in zoo-housed giraffes.

| Variable | Variable | | | | | | | | |
|---|---|---|---|---|---|---|---|---|---|
| | Age | BCS | Cortisol | DHEA-S | Cortisol/DHEA-S | Cholesterol | NEFA | Fructosamine | Allostatic Load |
| Age | | 0.30 | −0.26 | −0.41 | 0.04 | 0.33 | 0.10 | 0.42 | 0.10 |
| | | NS | NS | 0.09 | NS | NS | NS | 0.09 | NS |
| BCS | 0.30 | | −0.09 | −0.18 | 0.09 | 0.00 | 0.32 | −0.23 | −0.41 |
| | NS | | NS | NS | NS | NS | NS | NS | 0.09 |
| Allostatic Load | 0.10 | −0.41 | 0.29 | −0.14 | 0.38 | 0.33 | 0.07 | 0.68 | |
| | NS | 0.09 | NS | NS | NS | NS | NS | <0.01 | |
| Immobilizations | 0.67 | −0.09 | 0.00 | −0.31 | 0.16 | 0.27 | 0.29 | 0.48 | 0.35 |
| | <0.01 | NS | NS | NS | NS | NS | NS | 0.05 | NS |
| Illnesses | 0.40 | 0.23 | −0.23 | −0.58 | 0.16 | 0.10 | 0.31 | 0.16 | 0.16 |
| | 0.10 | NS | NS | 0.01 | NS | NS | NS | NS | NS |
| Pregnancies | 0.54 | 0.27 | −0.05 | −0.39 | 0.27 | −0.16 | 0.23 | 0.31 | 0.03 |
| | 0.02 | NS | NS | 0.10 | NS | NS | NS | NS | NS |
| Transports | 0.09 | 0.31 | 0.07 | −0.18 | 0.31 | −0.03 | 0.38 | −0.19 | 0.24 |
| | NS | NS | NS | NS | NS | NS | NS | NS | NS |
| Total Stress Events | 0.77 | 0.05 | −0.12 | −0.38 | 0.09 | 0.23 | 0.39 | 0.45 | 0.26 |
| | <0.01 | NS | NS | NS | NS | NS | NS | 0.06 | NS |
| Stress Events/Year | 0.52 | 0.37 | −0.39 | 0.01 | −0.25 | 0.20 | −0.32 | 0.18 | −0.28 |
| | 0.03 | NS | 0.10 | NS | NS | NS | NS | NS | NS |

[1] Spearman correlation coefficients (r) are the top number for each cell. *p*-values for each correlation are italicized and below each coefficient. BCS, body condition score; DHEA-S, dehydro-epiandrosterone sulfate; NEFA, non-esterified fatty acids; NS, not significant.

## 4. Discussion

In this study, we found that the accumulation of stress events encountered by zoo-housed giraffes over time was reasonably reflected in certain stress-responsive serum biomarkers and in the composite allostatic load index calculated from these indicators. Interestingly, higher values for allostatic load index were more closely related to the total

cumulative number of stress events experienced prior to the time of allostatic assessment than to the yearly frequency of stress events. In other words, a greater number of stress events coincided with higher allostatic load scores regardless of the number of years over which the events occurred. Among our initial panel of biomarkers, serum cortisol and fructosamine appeared to be most predictive of a high number of total stress events, as they were 65% and 49% greater, respectively, in giraffes from the highest 50th percentile for total stress events. Serum DHEA-S was also moderately correlated with the total number of stress events encountered. Nevertheless, the allostatic load index calculated from the six prospectively-selected stress biomarkers was almost 4-fold greater for giraffes from the highest 50th percentile for total stress events. Although refinement of the biomarker panel for captive giraffes appears warranted, these findings nonetheless demonstrate the marked benefits that the summative nature of allostatic load indices offer regarding the effective and reliable identification of long-term stress status. Furthermore, they demonstrate the importance of maintaining detailed medical and husbandry records for zoo-housed animals and illustrate the advantage of having these animals trained for voluntary blood draws.

The allostatic load index calculated for this group of zoo-housed giraffes indicates that the total number of accumulated stressful experiences is more influential than the timing or nature of the event. Indeed, allostatic load index values failed to correlate with any individual stress category but correlated well with total stress events. This supports previous postulations that the overall number of times that the stress response is activated is most associated with an animal's coping ability and longevity, as stress responses can be modulated substantially by previous experiences [3]. Of course, the timing and nature of stressful event experiences over a lifetime is not uniform across a population. For example, a 2-year-old giraffe in the present study had experienced seven documented stress events, whereas a 13-year-old giraffe had experience only six events. Likewise, the ability to cope with stress varies among individuals, as demonstrated by a 2-year-old giraffe with an allostatic load value of 4. Moreover, predisposing background factors such as demographics, genetics, and management can influence sensitivity to stress and, in turn, how allostatic load accumulates [9]. For all of these reasons, we expected to find a strong effect from stress event frequency. However, no such effects were observed, which further indicates that overall stress accumulation is the primary determinant allostatic ability and that it matters less how the stress arises.

Although the allostatic load index did not correlate more strongly to any one stress category than to others, some biomarkers did appear to correspond more closely with specific stressors. Pearson and Spearman correlations indicated that decreasing concentrations of DHEA-S coincided with increasing instances of pregnancies and illnesses. Low DHEA-S production could be reflective of adrenal fatigue, as the adrenal cortex is the primary site of synthesis [50]. Lower circulating concentrations would also be consistent with reduced capacity to cope with stress, as DHEA-S is believed to be protective against negative physiological outcomes of stress [20,51]. Elevated circulating cortisol was observed in giraffes with high overall stress event exposure, but concentrations did not correlate well with any specific stress categories. However, cortisol/DHEA-S ratios were not only strongly predictive of allostatic load and total stress events, but they were also correlated with the number of illnesses and transport events experienced. Elevated ratios of catabolic cortisol to anabolic DHEA-S have been implicated in poor immune function, anxiety, chronic illness, and other sources of chronic stress in humans and animals, as recently reviewed in detail [52]. The reliability of this ratio in predicting poor outcomes is likely because prolonged glucocorticoid exposure depletes circulating DHEA-S concentrations, which in essence doubles the negative effect on coping ability. This has been shown to manifest in cognitive and reproductive disorders and in poor immune responses [13,51,53]. Interestingly, the metabolic marker fructosamine was influenced by total immobilization occurrences in addition to allostatic load and total stress events. Increased fructosamine concentrations are often associated with stress-induced insulin resistance [54] and hyperglycemia in animals [55,56]. The reason for the stronger association of fructosamine to

immobilizations is not immediately clear, but the excitable nature of immobilization events may have played a role. Blood lipids are commonly used as biomarkers for cardiovascular, endocrine, and gastric disorders [31,57,58]. However, this study did not indicate that cholesterol or NEFA concentrations were strongly impacted by specific or cumulative stress events in the overall giraffe population. Supplementary analyses indicate that serum NEFA concentrations were moderately correlated with allostatic load and total stress events in females only, but further analyses indicated that NEFA concentrations were not different between males and females overall. Additionally, no differences in body condition score were observed between groups divided by total stress events, stress event frequency, age, or sex, although a modest negative Spearman correlation was observed with allostatic load. This is understandable, as BCS is a non-invasive health indicator that can typically be easily managed in captive environments through nutritional supplementation, exercise activity, and preventative medical care [46]. The differences in the relationships between individual index markers and specific stress categories further highlight the benefit of the allostatic load index for assessing long-term stress status, as it relies more on the severity of stress and stress responses rather than on their specific nature, which can be missed by individual biomarkers.

The age of the giraffes in this zoo-housed population had robust effects on several components of this study, which was not surprising. In general, older giraffes had lower circulating cortisol and DHEA-S but modestly higher cholesterol. Moreover, age correlated strongly with the frequency of stress events as well as the total number of events. The moderate negative correlation of age with serum DHEA-S reflects the natural decline that has been documented after adolescence in multiple mammalian species [51,59]. In humans, prepubertal DHEA concentrations are quite low [60,61]. However, this was not the case for giraffes in this study, as animals that were young enough to be prepubertal did not exhibit particularly low serum DHEA-S concentrations. It should be noted that the relationship between age and the number of specific and total stress events was of course not unexpected, as animals of greater age will have presumably had proportionally more opportunities to experience transfers, illnesses, pregnancies, and immobilizations. Importantly, however, this is not the case for allostatic load index, which did not differ between age groups or correlate with age.

## 5. Conclusions

The findings of this study lead us to conclude that allostatic load index is an effective approach in assessing chronic stress status in captive giraffe populations. Although future refinements of the biomarker panel used for this index will improve its effectiveness, the present study demonstrates conclusively that the index calculated from the composite biomarker indices better reflected the cumulative total of documented stress events experienced by the individual giraffe than did any individual biomarker. Because of the well-documented role that stress plays in morbidity and mortality risks, it is necessary to assess stress in a way that is independent of the specific details of the stress event and instead reflects the individual animal's physiological responsiveness to the event. Therefore, better predicting health needs and outcomes by more accurately assessing stress status using allostatic load indices may provide greater opportunities to improve welfare in captive giraffe populations.

**Supplementary Materials:** The following supporting information can be downloaded at: https://www.mdpi.com/article/10.3390/jzbg4030044/s1, Table S1: Pearson correlation coefficients[1] for stress biomarkers and occurrences of documented stress events in male zoo-housed giraffes; Table S2: Pearson correlation coefficients[1] for stress biomarkers and occurrences of documented stress events in female zoo-housed giraffes; Table S3: Pearson correlation coefficients[1] for stress biomarkers and occurrences of documented stress events in young (<10 years of age) zoo-housed giraffes; Table S4: Pearson correlation coefficients[1] for stress biomarkers and occurrences of documented stress events in older (10+ years of age) zoo-housed giraffes; Table S5: Spearman correlation coefficients[1] for stress biomarkers and occurrences of documented stress events in male zoo-housed giraffes; Table S6:

Spearman correlation coefficients[1] for stress biomarkers and occurrences of documented stress events in female zoo-housed giraffes; Table S7: Spearman correlation coefficients[1] for stress biomarkers and occurrences of documented stress events in young (<10 years of age) zoo-housed giraffes; Table S8: Spearman correlation coefficients[1] for stress biomarkers and occurrences of documented stress events in older (10+ years of age) zoo-housed giraffes.

**Author Contributions:** Conceptualization, H.N.B. and T.C.S.; methodology H.N.B. and T.C.S.; software, D.T.Y.; validation T.C.S., D.T.Y. and L.K.K.; formal analysis, H.N.B., T.C.S. and D.T.Y.; investigation, H.N.B., T.C.S. and D.T.Y.; resources, T.C.S. and L.K.K.; data curation, H.N.B., T.C.S. and D.T.Y.; writing—original draft preparation, H.N.B.; writing—review and editing, D.T.Y., T.C.S. and L.K.K.; visualization, H.N.B. and D.T.Y.; supervision, L.K.K. and T.C.S.; project administration, L.K.K. and T.C.S.; funding acquisition, T.C.S. and L.K.K. All authors have read and agreed to the published version of the manuscript.

**Funding:** This research was privately funded by the Lincoln Children's Zoo and Dr. Coleen Stice, MD, FACS, CPE.

**Institutional Review Board Statement:** The study was reviewed and conducted in accordance with Omaha's Henry Doorly Zoo and Aquarium's IACUC. Blood samples were collected by veterinary staff during medical procedures unrelated to this research project.

**Data Availability Statement:** The data presented in this study are available on request from the corresponding author. Public access to the data is restricted because permission to access the private server where the information is stored is limited to individuals affiliated with zoo entities.

**Acknowledgments:** The authors acknowledge Doug Armstrong, Julie Napier, and Nicole Linefelter at the Omaha Henry Doorly Zoo and Aquarium for collecting, preserving, and dispensing samples for evaluation.

**Conflicts of Interest:** The authors declare no conflict of interest. This publication does not necessarily represent the official views of any of these entities or individuals.

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
