# Peer review of "Allostatic Load Index Effectively Measures Chronic Stress Status in Zoo-Housed Giraffes"

_2673-5636, doi:10.3390/jzbg4030044_

Round 1

Reviewer 1 Report

Authors characterized the efficacy of using indicators of allostatic load on physiological measures of giraffe welfare, in relation to the number and frequency of standard stressful events in a zoo. The systems that react to stress are important protectors of the body in the short run, but cause damage and accelerate disease when chronically active. Therefore, the ideas and intention behind the project are important to understanding the pathophysiological responses of animals to human care, which is varied, multimodal, species specific, and therefore necessitates further studies like this.

Overall, the manuscript is well written, methods employed were appropriate for a condensed, single-institutional physiological study (i.e., outputs), however more information regarding sample collection and the environmental conditions (i.e., inputs) is needed to better understand and interpret the results more clearly. Overall, the statistical analysis were appropriate and well-explained in their assessment, and conclusions clearly drawn from the outcomes they received and by supported literature.

My comments are summarized below.

Lns 44-46; Lns 55-62. Recognizing this is an examination of these indicators as correlates of stress, which is still a burgeoning science with non-human animals given the breadth of taxa and their life histories, many of the citations are studies conducted with people. I recommend being more direct in either noting what is and is not known in non-human mammals, or in that the auspices of this study are to draw from human pathophysiology science in evaluating efficacy of primary mediators and secondary outcomes of ALI in animals.

Lns 47-49. If noting that this methodology has shown potential in animals, I recommend providing references or further a priori justifications for the claim.

Ln 75. Should this subheading be "Subjects"?

Ln 78. Recommend revising start of the sentence to be "Sample collections..." vs "Collections...", as collections in zoo-lingo generally refers to the animal residents, not procedure. I had to read the sentence a couple times to recognize it was referring to methods, not individuals.

Lns 81-82. Authors note that animals were of sound health and not on medications at time of sampling, however report (Table 1, Lns 147-154) need for immobilizations, illness, and pregnancies within the study period. Such events in giraffe are indicative of health issues which would question sound health or situations which would also require medications. Likewise, it is not uncommon for older giraffe (> 16yo in males, >20yo in females) to be on routine or periodic medication for managing arthritis, foot issues, or other common geriatric ailments. Please clarify.

Lns 82-83. How many samples were taken per individuals, and at what frequency within the years collected.

Sec. 2.6, ln 143. What period of time were stress events surveyed? And, how does this align with collection of samples? 

Reviewer 2 Report

This manuscript describes a novel application of the allostatic load index measurement as applied to 18 zoo-housed giraffe.  The manuscript is clearly written.  The introduction succinctly describes the need for an instrument to measure chronic stress in animals, and the results and conclusion accurately describe both the utility of this instrument and the need for further biomarker discovery.  While the number of subjects is a bit low, the finding that the allostatic load index was significantly different among animals with low vs high numbers of stress events is promising.  The use of two correlation coefficient calculations and the detailed tabular reporting of both is considered a bit overly confusing in this complex paradigm.  Perhaps it would be more understandable and less cumbersome to the readers to choose one of the correlation statistics and report it in full, while noting how it compared to the other test in particular areas. 

Some particular points from the text:

Line 70: 'volume and frequency' - is this meant to be 'NUMBER and frequency'"

Line 179: Here, serum NEFA concentrations were greater for giraffes in the highest 50th percential for stress event frequency, but in line 200, it is stated that serum NEFA tended to be negatively correlated with stress event frequency.  Is this correct?

Figure 5A.  Allostatic load is stated to be significantly higher in giraffes in the high total stress event group (line 189) but this is not reflected with an asterisk in this graph.

Line 347:  'panel used of this index' should be '...FOR this index.."

References:

1.  Cooper et al missing publisher information?

58.  Dadone missing book information (Fower's Zoo and Wild Animal Medicine vol. 10)

The quality of English in the manuscript is excellent.  Very few edits recommended, well written.
